# Self-Compassion and Anxiety in Adolescents with and without Anxiety Disorder

**DOI:** 10.3390/children10071181

**Published:** 2023-07-07

**Authors:** Edibe Tali, Eva S. Potharst, Esther I. de Bruin, Elisabeth M. W. J. Utens

**Affiliations:** 1Levvel, Rijksstraatweg 145, 1115 AP Duivendrecht, The Netherlands; e.tali@levvel.nl; 2UvA Minds, Academic Outpatient (Child and Adolescent) Treatment Centre, University of Amsterdam, Banstraat 29, 1071 JW Amsterdam, The Netherlands; epotharst@uvaminds.nl (E.S.P.); e.i.debruin@uva.nl (E.I.d.B.); 3Research Institute of Child Development and Education, University of Amsterdam, Nieuwe Achtergracht 127, 1018 WS Amsterdam, The Netherlands; 4Department of Child and Adolescent Psychiatry, Amsterdam University Medical Centers, Meibergdreef 9, 1105 AZ Amsterdam, The Netherlands; 5Department of Child and Adolescent Psychiatry/Psychology, Sophia Children’s Hospital, Erasmus Medical Center, Wytemaweg 8, 3015 CN Rotterdam, The Netherlands

**Keywords:** self-compassion, anxiety, anxiety disorder, adolescents, clinical

## Abstract

Previous studies have linked self-compassion to mental health, specifically anxiety, in non-clinical adolescents, suggesting that self-compassion can be a protective factor against anxiety. This study compared the overall level of self-compassion and (un)compassionate self-responding in adolescents with and without an anxiety disorder and assessed the association between self-compassion and anxiety. This cross-sectional study included adolescents (12–19 years) with an anxiety disorder (N = 23) and a reference group (N = 28). Participants completed the Self-Compassion Scale (SCS) and State Trait Anxiety Inventory (STAI). Results showed that overall self-compassion and uncompassionate self-responding were significantly lower and higher in the clinical than the reference group, respectively, while compassionate self-responding did not differ between groups. In the clinical group, only uncompassionate self-responding was significantly associated with higher anxiety. In the reference group, uncompassionate self-responding showed a significant positive association with anxiety, and compassionate self-responding showed a significant negative association with anxiety. Although the results suggest that low uncompassionate self-responding may buffer against anxiety, the role of compassionate and uncompassionate self-responding remains unclear. An alternative explanation is that the uncompassionate self-responding items measure the presence of psychopathology in adolescents with an anxiety disorder. More research on the construct validity of the SCS uncompassionate self-responding scale is needed.

## 1. Introduction

Anxiety disorders are characterized by an elevated sensitivity to threat, as observed across symptoms reporting, behavioral, cognitive, and physiological responding, and underlying neural systems [1]. Anxiety disorders are among the most common mental disorders in adolescents; international estimates indicate that 5% of youth meet the criteria for any anxiety disorder at a given time [2]. Anxiety disorders among youth impede family, psychosocial and academic functioning [3,4]. The chronicity and impairments associated with these disorders result in a substantial economic burden on society, with annual costs of up to Euro 74 billion for Europe [5]. Considering the above, interventions that are acceptable and effective for adolescents with an anxiety disorder are needed to promote psychological well-being and reduce societal costs. Several meta-analyses and systematic reviews have shown that cognitive behavioral therapy is the most widely used and evidence-based treatment of anxiety disorders in youth [6,7]. However, approximately one-third of youths with an anxiety disorder are not free from their primary diagnosis at posttreatment or long-term follow-up [7]. Therefore, identifying protective factors in adolescence that can buffer against the development of (more severe) anxiety and comorbid disorders and that can enhance treatment outcomes is clinically relevant. The current study aims to study a potentially protective factor that may offer the possibility for (improved) treatment of anxiety disorders in adolescents, namely self-compassion. 

Self-compassion is a psychological concept rooted in Buddhist philosophy and is considered the same as compassion towards others merely turned inward [8]. Derived from Buddhist conceptualizations, it can be defined as an awareness of suffering in oneself and a sense of concern about it, coupled with the desire and motivation to relieve it [9,10]. According to Neff [11], self-compassion entails three components: (1) being kind and supportive to oneself when struggling, rather than being harsh and judgemental (self-kindness as opposed to self-judgment); (2) recognizing that struggling is a shared part of all human experience rather than feeling isolated from other people as a result of self-suffering (common humanity as opposed to isolation); and (3) maintaining a balanced perspective amid personal suffering rather than becoming absorbed by it (mindfulness as opposed to over-identification with struggling). The components of self-compassion may serve as an antidote to psychological processes that are characteristic of mental health problems, namely self-criticism, excessive self-control, and self-imposed rigid standards [12]. Low self-compassion may be related to decreased emotion regulation, which may be an important transdiagnostic factor underlying mental health problems [13]. In the last two decades, there has been an increasing amount of academic research on self-compassion as a factor that may buffer against mental health problems in general and anxiety disorders specifically [14].

For adults, meta-analyses in community samples have shown that self-compassion is negatively related to anxiety, depression, and stress [15] while being positively related to emotional well-being [16]. In an experimental laboratory setting with adults, self-compassion seemed to buffer against anxiety when faced with an ego threat [17]. Hoge et al. [18] found that all six components of self-compassion were lower in adults with generalized anxiety disorder compared to healthy controls, and self-compassion was negatively related to measurements of generalized anxiety disorder. In another study, self-compassion was lower in adults with social anxiety disorders [19]. In the same study, no correlation was found between self-compassion and two different measures of anxiety [19]. A recent meta-analysis on self-compassion interventions, including 65 randomized controlled trials, showed that self-compassion interventions had small to medium effects on anxiety, depression, and stress at post-test but that the effect on anxiety was no longer significant at follow-up [20].

Self-compassion may be especially relevant for adolescents because of the developmental tasks and associated stressors that are faced by adolescents in, for example, the school and social environment. An important developmental task is forming their own identity, gaining autonomy (more independency from parents), and belonging to the peer group, which is often accompanied by a tendency to continuously evaluate and compare themselves to others [21]. Especially when these evaluations and comparisons have a negative and self-critical tone, adolescents are at risk for anxiety and other mental health problems [22,23]. Until now, few studies have been conducted on self-compassion during adolescence. As in research with adults, in adolescent non-clinical populations, self-compassion was inversely associated with anxiety, depression [24,25], and stress [24,26]. It was also found that self-compassion negatively predicted anxiety and moderated the relationship between stress and anxiety [24]. Self-compassion was negatively related to social anxiety and, moreover, predicted social anxiety over and above depression and anxiety symptoms [27]. Self-compassion in adolescents is positively associated with resilience [28], life satisfaction [29], and emotional well-being [30,31]. Further, a meta-analysis (with youth between 10–19 years) showed that age seems to moderate the relationship between self-compassion and psychological distress outcomes: the strength of this relationship reduced with increased age [8]. In addition, intervention studies in healthy adolescent samples showed that cultivating self-compassion may be a valuable protective factor during adolescence. After a six-week course of mindful self-compassion training, self-compassion improved significantly and predicted a reduction in anxiety and perceived stress and increased life satisfaction [32]. In a short intensive meditation retreat for adolescents with elevated stress levels, increases in self-compassion predicted decreases in perceived stress, rumination, depressive symptoms, and negative affect, and also increases in positive affect and life satisfaction [33].

Most studies on self-compassion have used the Self-Compassion Scale (SCS) [11]. The SCS is a multidimensional questionnaire [34] consisting of six subscales: self-kindness, common humanity, mindfulness, self-judgment, isolation, and over-identification. The items of the first three subscales are positively formulated, and the items of the last three subscales are negatively formulated. The total score involves the scores of the positively formulated items combined with the reverse scoring of the negatively formulated items. There has been some controversy about the factor structure of the SCS. On the one hand, research shows evidence for six correlated factors with a single general factor, supporting the use of the total score representing self-compassion and the use of the six subscale scores [34,35]. On the other hand, several studies have demonstrated two factors based on the positively and negatively formulated subscales, which are referred to as compassionate and uncompassionate self-responding, respectively [36,37,38]. The items of the subscales self-judgment, isolation, and over-identification, which are negatively formulated, assess characteristics that are associated with psychopathology [39] and show a strong correlation with psychopathology, which might result in an inflated negative relationship between self-compassion and psychopathology [36,39,40]. In line with this, a meta-analysis with mostly adults showed that the negative subscales of the SCS were stronger related to psychopathology than the positive subscales [40]. In studies focusing on anxiety in both adults and children, it was also found that anxiety showed a stronger association with uncompassionate self-responding than with compassionate self-responding [19,38]. Muris et al. [38] argue that especially compassionate self-responding seems to measure the protective nature of self-compassion, whereas Neff [34] argues that a self-compassionate mindset is represented by increased compassionate self-responding and reduced uncompassionate self-responding at the same time (with the items of compassionate self-responding representing protection against psychopathology and the items of uncompassionate self-responding representing vulnerability for psychopathology). 

Despite the fact that self-compassion has been suggested to be a possible protective factor against anxiety in adolescents, self-compassion has not been studied yet in a clinical sample of adolescents with anxiety disorders. To increase understanding of such a protective influence and on possible differences between compassionate and uncompassionate self-responding, we investigated the levels of overall self-compassion, compassionate self-responding, and uncompassionate self-responding in a clinical population of adolescents with an anxiety disorder compared to adolescents from the general population, and studied the relation between anxiety and overall self-compassion and compassionate and uncompassionate self-responding. Results could also contribute to the scientific discussion on the role of uncompassionate self-responding and shed light on the question of whether uncompassionate and compassionate self-responding are two sides of the same coin or if only compassionate self-responding represents the protective nature of self-compassion. Also, the results of our study could inform the adaptation of existing interventions. We hypothesized that the level of self-compassion would be lower for adolescents with an anxiety disorder than for adolescents from the general population (without an anxiety disorder) and that the difference between the groups would be larger for uncompassionate self-responding than for compassionate self-responding. Also, we expected that for both groups of adolescents, higher levels of self-compassion would be associated with lower levels of anxiety. More specifically, we expected uncompassionate self-responding to have a stronger relationship with anxiety symptoms than compassionate self-responding. 

## 2. Materials and Methods

### 2.1. Study Design

The current study had a cross-sectional design. Participants from a clinical and a reference sample completed questionnaires on self-compassion and anxiety at one measurement occasion. The study objectives were to compare the level of self-compassion (compassionate and uncompassionate self-responding) between the clinical and the reference group and the study the association between self-compassion (uncompassionate and compassionate self-responding) in both the clinical and the reference group. 

### 2.2. Participants

Two groups of adolescents aged 12 to 19 years old participated in the current study, a clinical sample and a reference sample. The clinical sample was recruited from two academic mental health treatment centers for children and adolescents, both located in a large city with a mainly metropolitan population. Inclusion criteria for the clinical sample were a classification of anxiety disorder as the primary diagnosis (classified by the Diagnostic and Statistical Manual of Mental Disorders 5, DSM-5 [41]). Exclusion criteria were an estimated IQ below 85 and a classification of autism, as autism would be the primary diagnosis when anxiety disorder and autism would be diagnosed both. The reference sample was recruited from two regular high schools in the same city as the mental health institutions. The inclusion criterion was an education level of at least pre-vocational secondary education, and the exclusion criterion was treatment for anxiety symptoms in the last 6 months. 

Sixty-two adolescents gave informed consent, of which eleven participants were excluded from the analysis because of missing data (seven did not complete (part of) the anxiety questionnaire, one did not finish the self-compassion questionnaire, and three did not complete both questionnaires). A total of 51 adolescents participated in the current study, aged 12 to 19 (M = 15.26, SD = 2.05). The clinical sample consisted of 23 adolescents, and the reference group consisted of 28 adolescents. The sociodemographic and diagnostic characteristics are displayed in Table 1. 

### 2.3. Procedure

Upon approval by the medical ethical committee and the ethical review board, data collection was started at the first and second participating mental health treatment centers. Informed consent was obtained from all the participants themselves and also from their caretakers when aged between 12 and 16. Adolescents with an anxiety disorder were recruited via an information letter and a consent form given by their therapist. After providing consent, adolescents completed the questionnaires at home. Adolescents from the high schools were recruited at one of the schools by an information letter from one of the researchers who provided oral information about the study in class. At the other school, an email was sent by the school manager to adolescents and their parents with an information letter. From the start of the study in February 2019 until June 2020, the procedure described above was followed. Due to the COVID-19 pandemic, recruitment was completed digitally from June 2020 to October 2020 using the same materials and the same (digitally provided) questionnaires.

### 2.4. Measures

*Demographic and treatment-related characteristics*. The age and gender of the adolescents were assessed with two additional questions in the survey. For the adolescents with an anxiety disorder at one of the mental health treatment centers, additional information about the type of anxiety disorder and comorbid classifications was obtained with additional questions on the survey. For the adolescents of the other mental institution, this information was retrieved from the medical file by one of the researchers. The adolescents from the reference group also completed an extra item on the self-compassion questionnaire, asking whether they had received treatment for anxiety symptoms in the last 6 months.

*Self-compassion* was assessed with the Dutch version (SCS-NL [42]) of the Self-Compassion Scale [11], consisting of 24 items (response options on a 7-point Likert scale: 1 = ‘rarely or never’ to 7 = for ‘almost always’). In this Dutch version, two items from the original English version were omitted due to translation problems. Like the original version, the Dutch SCS-NL consists of six subscales: self-kindness, common humanity, mindfulness, self-judgment, isolation, and over-identification. The items in the first three subscales are positively formulated (and refer to compassionate self-responding), and the items in the latter three are negatively formulated (and refer to uncompassionate self-responding). The Dutch SCS-NL consists of a 7-point Likert scale, whereas the original version consists of a 5-point Likert scale. In the current study, a total score of the SCS was used with the negatively formulated items inversely scored [34]. Further, a compassionate self-responding score and an uncompassionate self-responding score were used without inversely scoring the negatively formulated items [43]. The SCS has been shown to have good reliability [11]. Cronbach’s alphas for the total sample in this study were 0.92 for the total scale, 0.81 for compassionate self-responding, and 0.93 for uncompassionate self-responding.

*Anxiety* was assessed using the Dutch version of the State Trait Anxiety Inventory for children (STAI-DY [44]; STAI-C [45]. The STAI-DY consists of 40 items, administered on a 4-point Likert scale: 20 items assessing state anxiety (1 = ‘not at all’ to 4 = ‘often’) and 20 items assessing dispositional, trait anxiety (1 = ‘almost never’ to 4 = ‘almost always’). The Dutch version has good validity and reliability, with a Cronbach’s alpha between 0.90 and 0.93 [44]. In this study, Cronbach’s alpha for the total sample was 0.98 for the total scale of the questionnaire.

### 2.5. Data Analysis

SPSS 27 was used for all statistical analyses. Inspection of variable distributions indicated sufficient normality: skewness and kurtosis of all variables were <|1.5|. An analysis of standard residuals showed that the data contained no outliers (<3). The variables met the assumption of non-zero variances (see standard deviations in Table 2) and of homogeneity of variance, as shown by non-significant (*p* > 0.05) Levene’s test. Scatter plots of combinations of independent and dependent variables showed that data met the assumption of linearity. Multicollinearity was checked by checking the VIF and Tolerance values of the regression analyses (VIFs < 10, Tolerance > 0.1). To check for differences between the clinical group and reference group regarding age and gender, an independent *t*-test and a Chi-square test were conducted, respectively. Pearson’s correlations were calculated to test whether anxiety, overall self-compassion, and compassionate and uncompassionate self-responding were associated in both the clinical and the reference groups.

To test whether the groups differed on overall self-compassion, compassionate self-responding, and uncompassionate self-responding, while taking into account adolescents’ age, multiple linear regression analysis was performed. In these analyses, the groups and adolescents’ age were used as independent variables, and overall self-compassion, compassionate self-responding, and uncompassionate self-responding, respectively, as the dependent variable. 

Multiple linear regression analyses were also performed to test whether the level of self-compassion was associated with the level of anxiety. These analyses were done for the clinical group and the reference group separately. We first tested whether overall self-compassion was a significant predictor of anxiety while taking age into account by using age and overall self-compassion as independent variables and anxiety as the dependent variable. The backward method was used for this regression analysis, which begins by placing the predictors in the model and, at each step, gradually eliminating predictors from the model that make the least contribution to the model (using *p* > 0.05 as the removal criterion) to find a reduced model explaining the data best [46]. In the clinical group, age was removed as an independent variable in model 2, while, in the reference group, age was kept in the model. To explore the contribution of compassionate and/or uncompassionate self-responding on anxiety, we again used multiple linear regression analyses for the clinical and the reference group separately. In these analyses, compassionate and uncompassionate self-responding were used as independent variables, anxiety was used as the dependent variable, and the backward method was used to eliminate variables that did not contribute to the model. In the clinical group, compassionate self-responding was removed as an independent variable in model 2, while in the reference group, both compassionate and uncompassionate self-responding were kept in the model. 

## 3. Results

### 3.1. Initial Analyses

An independent *t*-test showed that the participants in the clinical group (*M* = 16.1; *SD* = 1.89) were significantly older than the participants in the reference group (*M* = 14.6; *SD* = 1.97), *t* (49) = 2.69, *p* = 0.01, *d* = 0.75. A Chi-square test showed there was no difference in gender between the groups, *X*² (1, *n* = 51) = 0.40, *p* = 0.69. The means and standard deviations of all measures for the clinical and reference group are shown in Table 2. Correlations among the variables self-compassion and anxiety for both groups are presented in Table 3 and show significant correlations between all measures for both groups.

### 3.2. Self-Compassion: Clinical versus Reference Group

Consistent with our first hypothesis, the adolescents in the clinical group had significantly lower (total) self-compassion scores than those in the reference group, *F* (2, 48) = 7.63, *p* = 0.001. Age was not significantly associated with self-compassion (*t* (50) = −1.70, *ns*, *β =* −0.23). When the analyses were conducted for compassionate and uncompassionate self-responding (the positive and negative subscales, respectively), there was no difference in compassionate self-responding between the two groups when age was taken into account (*F* (2, 48) = 2.58, *ns*.) The groups did differ on uncompassionate self-responding, with adolescents in the clinical group reporting significantly higher concerning uncompassionate self-responding than adolescents in the reference group, *F* (2, 48) = 8.97, *p* = 0.000. Age was not significantly associated with uncompassionate self-responding (*t* (50) = 1.34, *ns*, *β =* 0.18).

### 3.3. Self-Compassion and Anxiety in the Clinical Group

Further, multiple regression analyses showed that the level of self-compassion was negatively related to the level of anxiety for the clinical group. As can be seen in Table 4, a significant regression equation was found for self-compassion, where age was deleted from the model since it was not significantly associated with anxiety. Level of self-compassion was associated with anxiety, predicting 59.7% of the variance in anxiety. To explore the contribution of compassionate and uncompassionate self-responding to anxiety, another multiple regression analysis with the backward method was carried out. A significant regression equation was found, where compassionate self-responding was deleted from the model since it was not significantly associated with anxiety. Uncompassionate self-responding was positively associated with anxiety, predicting 73.7% of the variance in anxiety.

### 3.4. Self-Compassion and Anxiety in the Reference Group

Finally, regression analyses showed that within the reference group, the level of self-compassion was also negatively related to anxiety. As shown in Table 5, a significant regression equation was found, where age significantly contributed to the model. Level of self-compassion and age together predicted 58.6% of the variance in anxiety. Another multiple regression analysis was carried out to explore the contribution of compassionate and uncompassionate self-responding to anxiety. A significant regression equation was found where compassionate self-responding was negatively related to anxiety, and uncompassionate self-responding was positively related to anxiety, predicting 49.8% of the variance in anxiety.

### 3.5. Post-Hoc Power Calculations

A post hoc power analysis was conducted using the software G*Power [47]. The 2-variables models were used, and an alpha error probability of 0.05 was selected. For the regression analysis testing the difference between the groups (*n* = 51), in which an effect size of ƒ² = 0.32 was shown, we found a power of 0.95. For the regression analyses testing the association between self-compassion and anxiety in the clinical group (*n* = 23), in which effect sizes of ƒ² > 1.63 were shown, the power was found to be >0.99. For the regression analyses testing the association between self-compassion and anxiety in the reference group (*n* = 28), in which effect sizes of ƒ² > 0.99 were shown, the power was found to be >0.99.

## 4. Discussion

### 4.1. Principal Findings

The purposes of the present study were to compare the level of self-compassion of adolescents with an anxiety disorder versus that of adolescents from the general population and to study the relationship between self-compassion and anxiety in both groups. As was expected, we found that adolescents with an anxiety disorder scored lower on overall self-compassion. We expected to find a larger difference between the groups on uncompassionate self-responding than on compassionate self-responding. We indeed found a large difference in uncompassionate self-responding, with the anxious group showing more uncompassionate self-responding than the reference group and no significant difference in compassionate self-responding. Furthermore, we expected that for both groups, lower self-compassion would be associated with higher anxiety and that uncompassionate self-responding would show a stronger relationship to anxiety than compassionate self-responding. Within the clinical group, we found an association (large effect) between self-compassion and anxiety, which was explained completely by uncompassionate self-responding and not by compassionate self-responding. Within the reference group, both compassionate and uncompassionate self-responding explained variance in the prediction of anxiety. 

### 4.2. Self-Compassion in Adolescents with and without an Anxiety Disorder

The finding that self-compassion was lower in the group of adolescents with an anxiety disorder than in the group of adolescents without such a disorder corresponds with previous research showing this same pattern when comparing adults with social anxiety disorder and generalized anxiety disorder with healthy controls as to self-compassion [18,19]. Previous studies in adolescents did not yet include clinical samples of adolescents with an anxiety disorder but did study associations between symptoms of anxiety and self-compassion [24,25,27]. The finding from the current study that self-compassion was associated with anxiety is consistent with the results of these previous studies in adolescents [24,25,27]. An explanation of the combination of these findings could be that a high level of self-compassion may protect adolescents against anxiety. However, it should be noted that all these studies, including the current one, have a cross-sectional study design, limiting conclusions about the directionality of the associations found [13]. A study in adults on self-compassion, mindfulness, and anxiety did use a longitudinal design and cross-lagged statistical models that did show the temporal ordering of changes in self-compassion, mindfulness, and anxiety [48]. Interestingly, while self-compassion (and not mindfulness) showed the strongest correlation with anxiety on the same measurement occasion, mindfulness (and not self-compassion) was a significant predictor of anxiety at a later time point [48]. The results of that longitudinal study emphasize the importance of being careful in conclusions that may be drawn based on the results of the current study. Although we did find that adolescents with an anxiety disorder have lower self-compassion, and high anxiety was associated with low self-compassion, it is not necessarily so that low self-compassion is a working mechanism in the development of anxiety disorders. 

### 4.3. Compassionate and Uncompassionate Self-Responding

Concerning compassionate and uncompassionate self-responding, it was found that adolescents with an anxiety disorder experienced higher uncompassionate self-responding than non-anxious adolescents, whereas there was no difference between the groups on compassionate self-responding. When the relationship between these two components and anxiety was studied, the results were different for the clinical and the reference group. Although the positive association between uncompassionate self-responding and anxiety was greater than the negative association between compassionate self-responding and anxiety in the reference group, both were significant, which aligns with prior research [38]. However, in the clinical group, only uncompassionate self-responding was a significant (positive) predictor of anxiety. Previous research also showed that symptoms of psychopathology, such as anxiety and uncompassionate self-responding, are strongly related and that this relationship is stronger than the relationship between anxiety and compassionate self-responding [27,34,38]. Neff [34] argues that a self-compassionate mindset represents increased compassionate self-responding and reduced uncompassionate self-responding and that the stronger association between uncompassionate self-responding and psychopathology symptoms indicates reduced tendencies to be self-compassionate. In line with this reasoning, it would be expected that compassionate self-responding would be lower in adolescents with an anxiety disorder compared to non-anxious adolescents and that compassionate self-responding within the group of adolescents with an anxiety disorder would also have a significant relationship with anxiety, but this was not found. One possible explanation is that the SCS operates differently in a clinical population [49]. It could be that adolescents with an anxiety disorder respond differently to the SCS because they are more prone to negativity bias than non-anxious adolescents [50,51]. This might be reflected in a tendency for adolescents with an anxiety disorder to respond with more emphasis on the negative items of the SCS than on the positive ones, resulting in more uncompassionate than compassionate self-responding.

Another, and possibly related explanation is that since uncompassionate self-responding in non-clinical populations is more strongly related to symptoms of psychopathology, especially internalizing problems, than compassionate self-responding [34,43], it could be that the already existing strong association between uncompassionate self-responding and psychopathology is magnified in a population with clinical anxiety. Some of the items of the SCS that intend to measure uncompassionate self-responding do not only reflect uncompassionate ways that one might react to pain and suffering but also measure aspects of psychopathology itself [39]. The extremely large positive correlation of 0.866 between uncompassionate self-responding and anxiety in the clinical group seems to confirm the hypothesis that there may be an overlap between the negative items of the SCS and the anxiety measure used in this study. An example of items that may be influenced by the anxiety disorder of adolescents may be ‘When something upsets me, I get carried away with my feelings’ and ‘When I’m feeling down, I tend to obsess and fixate on everything that’s wrong’ (subscale over-identification). Being carried away by anxiety by, for example, having a panic attack, being self-absorbed, and ruminating, may be very familiar for these adolescents and characteristic of their form of psychopathology. A high score on the item ‘I’m intolerant and impatient towards those aspects of my personality I don’t like’ (subscale self-judgment) may merely reflect the level of suffering that adolescents experience because of the limitations that they experience in their lives due to the anxiety disorder, and an item like ‘When I am feeling down I tend to feel like most other people are probably happier than I am’ (subscale isolation) may be a quite realistic thought for adolescents that require specialized mental health care that most of their peers probably do not need. When the results of our study are seen in the light of the scientific discussion on the role of uncompassionate self-responding, the outcomes of our study do not seem to confirm the idea that uncompassionate and compassionate self-responding are two sides of the same coin, at least in a clinical sample of adolescents with an anxiety disorder. More research on the construct validity of the uncompassionate self-responding scale of the SCS in different clinical and non-clinical samples of different age groups, including adolescents, is still needed.

An alternative model of self-compassion to that of Neff [11] is offered by Gilbert in his social mentality theory of compassion [52]. He supposes that self-compassion can relate to the self through evolutionary systems, where self-compassion (compassionate self-responding) and self-criticism/self-coldness (uncompassionate self-responding) are associated with different systems. Self-compassion is associated with the warmth and safeness (parasympathetic) system, which concerns emotions, cognitions, and behaviors promoting positive relating to the self (e.g., self-soothing and self-compassion) and is also supposed to deactivate defensive emotions and behaviors, such as anxiety and flight responses. Self-criticism is associated with the threat–defense (sympathetic system), which concerns emotions, cognitions, and behaviors aimed at reducing threat (e.g., being self-critical and anxious). Gilbert et al. [53] assume that the positive and negative items of the SCS should not be used to measure self-compassion as a unitary construct since positive and negative affect are generally seen as independent dimensions of effect. 

### 4.4. Practical Implications 

Although there is still uncertainty about whether compassionate and uncompassionate self-responding act as buffers against (the development of) an anxiety disorder, there is some evidence that compassion-based interventions could make a valuable contribution to the treatment of adolescents with an anxiety disorder. A meta-analysis of the effects of various compassion-based interventions showed that after a compassion-based intervention, self-compassion and well-being improved significantly in adults, whereas anxiety and other symptoms of psychopathology decreased significantly [54]. In a review on self-compassion as an active ingredient in the prevention and treatment of anxiety and depression in young people (14–24 years), it was concluded that there is evidence for self-compassion intervention in decreasing anxiety and depression [55]. A qualitative study was also included in this review, for which both four self-compassion experts and 20 young people were interviewed. Self-compassion experts underscored the importance of decreasing self-criticism, and young people confirmed this by saying that they would be more interested in a treatment aimed at reducing self-criticism than in a treatment aimed at improving self-compassion [55]. The concept of self-criticism is related to the concept of self-judgment, one of the aspects of uncompassionate self-responding. As adolescents with and without an anxiety disorder only differed on uncompassionate self-responding and not on compassionate self-responding, it seems logical to support adolescents in decreasing uncompassionate self-responding and not necessarily increasing compassionate self-responding. However, just unlearning something is more difficult than learning an alternative. Therefore, it seems logical to teach adolescents with high levels of anxiety self-kindness as an alternative to self-judgment and self-criticism. 

### 4.5. Strengths, Limitations, and Future Directions

As previous studies did investigate the relationship between self-compassion and anxiety in adolescents but only included community samples, a strength of the current study was the inclusion of a clinical group of adolescents with an anxiety disorder. Another strength was including a reference group, with which the clinical group could be compared. 

This study was limited by several factors. This study had a cross-sectional design, making it impossible to determine the direction of effects or causality. While the statistical power of the present study was sufficient, it is possible that the sample sizes were too small to accurately generalize to larger groups and to show an additional contribution of compassionate self-responding on anxiety in the regression analyses for the clinical group. Because of the sample sizes, it was not possible to examine the six subscales separately. Second, the clinical group consisted of adolescents with differences in the primary diagnoses of anxiety disorders, raising concerns about the homogeneity of the group. Different forms of anxiety might be differently related to self-compassion in general and to compassionate and uncompassionate self-responding in particular. Further, the adolescents in the clinical group differed in the duration of treatment and received different kinds of treatment (e.g., inpatient and outpatient treatment). Adolescents receiving longer and more intense treatment could have experienced more self-compassion, particularly compassionate self-responding, due to the effects of treatment. Finally, the non-anxious adolescents were significantly younger than those with an anxiety disorder. There might be differences in self-compassion through different stages of development, and the SCS might operate differently for young and older adolescents. It is not unlikely that young adolescents, in general, might find the items of the SCS difficult to understand. Neff et al. [48] have developed a Self-Compassion Scale for Youth (SCS-Y) for adolescents between 10 and 14 years of age. 

Future research should address the limitations of this study by investigating larger samples. It may also be good to include more homogeneous groups concerning the duration and type of treatment, diagnoses, and age. The use of a scale for youth (SCS-Y) [56] is recommended for adolescents younger than 14 years of age. But most importantly, a longitudinal study design is recommended to shed more light on the directionality of the effects. A study design similar to the design chosen by Bergen-Cico and Cheon [48], in which mindfulness and compassionate and uncompassionate self-responding are included as mediators of change in anxiety, is recommended. Finally, an investigation of the construct validity of the uncompassionate self-responding subscales of the SCS in different clinical groups is needed. 

## 5. Conclusions

This study was the first to compare a clinical group of adolescents with anxiety disorders to adolescents from the general population, showing a difference in uncompassionate self-responding between adolescents between the groups, showing an association between uncompassionate self-responding and anxiety in adolescents with an anxiety disorder, and showing an association between compassionate and uncompassionate self-responding and anxiety in adolescents from the general population. The results of our study do not confirm the idea that compassionate and uncompassionate self-responding are two sides of the same coin. A practical implication of our study is that when self-compassion is added to treatment for adolescents with an anxiety disorder, it is important to help adolescents decrease uncompassionate self-responding, such as self-judgment. 

## Figures and Tables

**Table 1 children-10-01181-t001:** Sociodemographic and diagnostic characteristics of the participating adolescents.

	Clinical Group (N = 23)	Reference Group (N = 28)
Age (M ± SD)	16.07 ± 1.89	14.60 ± 1.97
Gender: girl	16 (69.6%)	18 (64.4%)
Level of education		
Pre-vocational secondary education	8 (34.8%)	0 (0%)
Senior general secondary education	8 (34.8%)	13 (46.4%)
Pre-university secondary education	6 (26.1%)	15 (53.6%)
Missing	1 (4.3%)	0 (0%)
Treatment		
Inpatient	6 (26.1%)	
Outpatient	17 (73.9%)	
Primary anxiety disorder		
Social anxiety disorder	11 (47.8%)	
Generalized anxiety disorder	6 (26.1%)	
Unspecified anxiety disorder	2 (8.7%)	
Panic disorder	1 (4.3%)	
Specific phobia	1 (4.3%)	
Separation anxiety	1 (4.3%)	
Anxiety disorder due to another medical condition	1 (4.3%)	
Number of comorbid diagnoses		
Zero	6 (26.1%)	
One	11 (47.8%)	
Two	4 (17.4%)	
Three	2 (8.7%)	
Comorbid diagnoses		
Another anxiety disorder	6 (26.1%)	
Depressive disorder	5 (21.7%)	
Obsessive-compulsive disorder	5 (21.7%)	
Post-traumatic stress disorder	4 (17.4%)	
Attention deficit hyperactivity disorder	3 (13.0%)	
Somatoform disorder	2 (8.6%)	

**Table 2 children-10-01181-t002:** Means, standard deviations, range of anxiety, and self-compassion measures for the clinical and reference group, while taking age into account.

	Clinical Group (N = 23)	Reference Group (N = 28)
**Anxiety**		
M ± SD	2.67 ± 0.65 ***	1.84 ± 0.48
Min–Max	1.08–3.55	1.05–2.80
**Overall self-compassion**		
M ± SD	3.29 ± 0.98 **	4.31 ± 0.89
Min–Max	1.58–5.63	3.08–6.67
*Compassionate self-responding*		
M ± SD	3.58 ± 1.00	4.05 ± 0.94
Min–Max	1.58–5.67	1.92–6.50
*Uncompassionate self-responding*		
M ± SD	4.99 ± 1.26 ***	3.43 ± 1.18
Min–Max	2.25–6.67	1.17–5.50

Compassionate Self-Responding represents the score of the subscales Self-kindness, Common humanity, Mindfulness (positive items), and Uncompassionate Self-Responding represents the score of subscales Self-judgement, Isolation, and Over-identification (negative items, without inversely scoring) of the Self-Compassion Scale. ** *p* < 0.01, *** *p* < 0.001.

**Table 3 children-10-01181-t003:** Pearson’s correlations of the independent variables (overall self-compassion, compassionate self-responding, and uncompassionate self-responding) and the dependent variable (anxiety) for the clinical and reference group.

	Anxiety		Compassionate Self-Responding	
	Clinical Group	Reference Group	Clinical Group	Reference Group
Overall self-compassion	−0.785 ***	−0.706 ***		
Compassionate self-responding	−0.426 *	−0.527 **		
Uncompassionate self-responding	0.866 **	0.637 ***	−0.490 **	−0.390 *

Compassionate self-responding represents the score of the subscales Self-kindness, Common humanity and Mindfulness (positive items), and Uncompassionate self-responding represents the score of subscales Self-judgement, Isolation and Over-identification (negative items, without inversely scoring) of the Self-Compassion Scale. * *p* < 0.05, ** *p* < 0.01, *** *p* < 0.001.

**Table 4 children-10-01181-t004:** Summary of the backward multiple linear regression models for anxiety in the clinical group.

		*β*	Adjusted R²	R²	ΔR²	F	ƒ²
Model 1	Overall self-compassion	−0.749 **	0.582	0.620	0.620	16.31 ***	1.63
	Age	0.075					
Model 2	Overall self-compassion	−0.785 ***	0.597	0.616	−0.004	33.62 ***	1.60
Model 1	Compassionate self-responding	−0.002	0.724	0.749	0.749	29.91 ***	3.00
	Uncompassionate self-responding	0.423 ***					
Model 2	Compassionate self-responding	0.423 ***	0.737	0.749	0.000	62.81 ***	3.00

For each analysis, two regression models were run using a backward deletion approach (criteria out *p* < 0.10). The second model is the selected model with one predictor. Compassionate Self-Responding represents the score of the subscales Self-kindness, Common humanity and Mindfulness (positive items) and Uncompassionate self-responding represents the score of subscales Self-judgement, Isolation and Over-identification (negative items, without inversely scoring) of the Self-compassion Scale. ** *p* < 0.01, *** *p* < 0.001.

**Table 5 children-10-01181-t005:** Summary of the backward multiple linear regression models for anxiety in the reference group.

		*β*	Adjusted R²	R²	ΔR²	F	ƒ²
Model 1	Overall self-compassion	−0.374 **	0.552	0.586	0.586	17.67 ***	1.42
	Age	0.071 *					
Model 1	Compassionate self-responding	−0.166 *	0.458	0.498	0.498	12.38 ***	0.99
	Uncompassionate self-responding	0.206 **					

For each analysis, two regression models were run using a backward deletion approach (criteria out *p* < 0.10). The second model is the selected model with one predictor. Compassionate self-responding represents the score of the subscales Self-kindness, Common humanity and Mindfulness (positive items) and Uncompassionate self-responding represents the score of subscales Self-judgement, Isolation and Over-identification (negative items, without inversely scoring) of the Self-Compassion Scale. * *p* < 0.05, ** *p* < 0.01, *** *p* < 0.001.

## Data Availability

The data may be available from the corresponding author, E.U., upon reasonable request and considering privacy legislation.

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
