# Peer review of "Self-Compassion and Anxiety in Adolescents with and without Anxiety Disorder"

_children, 2023, doi:10.3390/children10071181_

Round 1

Reviewer 1 Report

Manuscript Review Report

To,

Editor-in-chief,

Children

Dear Sir/Madam,

Thank you for inviting me to review the manuscript entitled "Self-Compassion and Anxiety in Adolescents with and without Anxiety Disorder" (Ms. ID: children-2480392) submitted to Children.

The study compares the overall level of self-compassion, and (un)compassionate self-responding in adolescents with an anxiety disorder to that of adolescents in the general population and assesses the association between self-compassion and anxiety. This study focuses on novel aspects of human behaviour. Below are my observations:

        (1)    The study title is good.

        (2)    The abstract needs minor changes. For example, I suggest avoid claiming a causal relationship between self-compassion and anxiety based only on a small study.

        (3)    The introduction also needs minor modifications. I suggest that the authors strengthen the study rationale by emphasizing the theoretical, practical, and policy implications. 

        (4)    The methods section is sufficient.  

        (5)    The results section is also good.

        (6)    The discussion part needs minor amendments. The interpretations of the study findings should be done in continuation with the research questions. Detailed theory, research and practice implications of the study results can be added. Novel findings should be highlighted. Some suggestions for future directions may be added by a subheading at the end of the discussion along with some limitations.

        (7)    The references are ok.

Essentially, the study is a good work and its findings may contribute to the area of ​​self-compassion and anxiety. Nevertheless, small changes are suggested. As it stands, the manuscript may be accepted after minor modifications as suggested above.

Thanks again to the editor for the opportunity to review the manuscript.

With best regards,

Reviewer

Author Response

Thank you for inviting me to review the manuscript entitled "Self-Compassion and Anxiety in Adolescents with and without Anxiety Disorder" (Ms. ID: children-2480392) submitted to Children.

The study compares the overall level of self-compassion, and (un)compassionate self-responding in adolescents with an anxiety disorder to that of adolescents in the general population and assesses the association between self-compassion and anxiety. This study focuses on novel aspects of human behaviour. Below are my observations:

  • Thank you for the time and effort you put into our manuscript.
  • The study title is good.
  • Thank you, we left the title unchanged.
  • The abstract needs minor changes. For example, I suggest avoid claiming a causal relationship between self-compassion and anxiety based only on a small study.
  • We agree with the reviewer that in the following sentence of the abstract, suggested a causal relationship: The results imply that especially low uncompassionate self-responding may buffer against symptoms of anxiety. Therefore, we replaced this sentence with the following sentence , now on page 1, line 27-29: “Although the results seem to suggest that low uncompassionate self-responding may buffer against anxiety in adolescents with an anxiety disorder, the role of compassionate and uncompassionate self-responding remains unclear.”
  • The introduction also needs minor modifications. I suggest that the authors strengthen the study rationale by emphasizing the theoretical, practical, and policy implications. 
  • Thank you for your suggestion. We agreed that it is important to emphasize theoretical and practical implications or four study. Because the small size of our study and the cross-sectional nature, we were, however hesitant to suggest that our study could have policy We adjusted the first half of the last paragraph of the Introduction on page 3 and 4, lines 139-152,  as follows: “
    Despite the fact that self-compassion has been suggested to be a possibly protective factor against anxiety in adolescents, self-compassion has not been studied yet in in a clinical sample of adolescents with anxiety disorders. To increase understanding on such a protective influence, and on possible differences between compassionate and uncompassionate self-responding, we investigated the levels of overall self-compassion, compassionate self-responding and uncompassionate self-responding in a clinical population of adolescents with an anxiety disorder compared to adolescents from the general population, and studied the relation between anxiety and overall self-compassion, compassionate and uncompassionate self-responding. Results could also contribute to the scientific discussion on the role of uncompassionate self-responding, and shed light on the question whether uncompassionate and compassionate self-responding are two sides of the same coin, or it is actually only compassionate self-responding represents the protective nature of self-compassion. Also, results of our study could inform the adaptation of existing interventions.”

        (4)    The methods section is sufficient.  

        (5)    The results section is also good.

  • Thank you for your positive feedback on the methods and results section.

        (6)    The discussion part needs minor amendments. The interpretations of the study findings should be done in continuation with the research questions. Detailed theory, research and practice implications of the study results can be added. Novel findings should be highlighted. Some suggestions for future directions may be added by a subheading at the end of the discussion along with some limitations.

  • We have rewritten a large part of the discussion, and gave it more structure by including subheadings. We started the discussion with a summary of the results on page 10 and 11, lines 421-437. Then we gave explanations for the study findings in continuation with the research questions on page 11 and 12, lines 438-489, followed by more detailed theory on page 12 and 13, lines 490-533. We made a separate paragraph on practice implications on page 13, lines 534-557. Research implications were included in the future directions, that we combined with the paragraph on the study strengths and limitations on page 13 and 14, lines 558-593. We hope that we were able to address your feedback in the revised version of the discussion.

        (7)    The references are ok.

Essentially, the study is a good work and its findings may contribute to the area of ​​self-compassion and anxiety. Nevertheless, small changes are suggested. As it stands, the manuscript may be accepted after minor modifications as suggested above.

Reviewer 2 Report

The study proposed by Tali E, et al. presents observations and suggestions for improving its writing and structure.

I will divide it into sections:

Abstract: The type of study proposed is not clearly shown, and the description of the methods used needs improvement. The results and conclusions are not clear in relation to the stated objectives.

Introduction: It is necessary to address the context and relevance of the topic clearly, as well as the definition and characteristics of anxiety disorder and self-compassion as a psychological construct. The authors should address these key points to create a coherent and clear text and establish the relationship between self-compassion and anxiety.

Methods: The wording and organization of the presented information need improvement. It is important to clarify the type of study conducted and provide information on the sample calculation. The description of the 51 participants should be structured in a table for a clearer and more appealing presentation to the reader. Additionally, the specific and main aspects related to the study objectives should be clearly and concisely written. The clarity of the data analysis should also be improved.

Results: The clarity of the main results needs improvement, and the terms and acronyms used should be reviewed.

Discussion: The authors adequately present a concise summary of the results, but it is necessary to provide plausible explanations for the observed results. The discussion should address the possible theoretical and practical implications of the results, as well as the limitations and strengths of the study and its impact on public health.

Conclusions: The conclusions should be modified according to the study objectives, avoiding including points that belong in the discussion section. They should focus on the strength of the study and its relevance in the research field.

English consistency, terms used, and grammar should be reviewed.

Author Response

The study proposed by Tali E, et al. presents observations and suggestions for improving its writing and structure.

  • Thank you for reviewing our manuscript and for your valuable comments.

Abstract: The type of study proposed is not clearly shown, and the description of the methods used needs improvement. The results and conclusions are not clear in relation to the stated objectives.

  • On the basis of your feedback, and given the maximum of 200 words, we have now adjusted the abstract on page 1, lines 16-32: “Previous studies have linked self-compassion to mental health, specifically anxiety, in non-clinical adolescents, suggesting that self-compassion can be a protective factor against anxiety. This study compared the overall level of self-compassion, and (un)compassionate self-responding in adolescents with and without an anxiety disorder, and assess the association between self-compassion and anxiety. This cross-sectional study included adolescents (12-19 years) with an anxiety disorder (N=23) and a reference group (N=28). Participants completed the Self-Compassion Scale (SCS) and State Trait Anxiety Inventory (STAI). Results showed that overall self-compassion and uncompassionate self-responding were significantly lower and higher in the clinical than the reference group, respectively, while compassionate self-responding did not show a difference between groups. In the clinical group, only uncompassionate self-responding was significantly associated with higher anxiety. In the reference group, uncompassionate and compassionate self-responding were significantly positively and negatively associated with anxiety, respectively. Although the results seem to suggest that low uncompassionate self-responding may buffer against anxiety, the role of compassionate and uncompassionate self-responding remains unclear. An alternative explanation is that the uncompassionate self-responding items measure the presence of psychopathology in adolescents with an anxiety disorder. More research on the construct validity of the SCS uncompassionate self-responding scale is needed.”

Introduction: It is necessary to address the context and relevance of the topic clearly, as well as the definition and characteristics of anxiety disorder and self-compassion as a psychological construct. The authors should address these key points to create a coherent and clear text and establish the relationship between self-compassion and anxiety.

  • Thank you for your suggestions regarding the introduction. We made the following changes to this section:
  • We added a definition of anxiety disorders at the beginning of the Introduction on page 1, line 36-38: “Anxiety disorders are characterized by an elevated sensitivity to threat, as observed across symptoms reporting, behavioural, cognitive and physiological responding, and underlying neural systems [1].”
  • We lengthened the first paragraph of the Introduction, and finished this first paragraph with a clarification of the aim of the study. We added the following text at page 2, lines 52-54: “The current study aims to study a potentially protective factor, that may potentially offer possibility for (improved) treatment of anxiety disorders in adolescents, namely self-compassion.”
  • The second paragraph of the Introduction at page 2, lines 55-65 is now started with the description, definition and characteristics of self-compassion.
  • To give some more explanation on the relationship between self-compassion and anxiety, we included the following two sentences to this second paragraph at page 2, lines 66-70: “The components of self-compassion may serve as an antidote to psychological processes that are characteristic of mental health problems, namely self-criticism, excessive self-control, and self-imposed rigid standards [12]. Low self-compassion may be related to decreased emotion regulation, which may be an important transdiagnostic factor underlying mental health problems [13].”
  • We included the paragraphs about associations between self-compassion and mental health in general and anxiety specifically in the text before moving on to a paragraph on the Self-Compassion Scale.
  • In the paragraph on self-compassion and anxiety in adolescents, the following text was added on the relevance for self-compassion in adolescence on page 2, line 86-93: “Self-compassion may be especially relevant for adolescents, because of the developmental tasks and associated stressors that are faced by adolescents, in for example the school and social environment. An important developmental task is forming of their own identity, gaining autonomy (more independency from parents), belonging to the peer group, which is often accompanied by a tendency to continuously evaluating and comparing themselves to others [21]. Especially when these evaluation and comparisons have a negative and self-critical tone, adolescents are at risk for anxiety and other mental health problems [22,23].”

Methods: The wording and organization of the presented information need improvement. It is important to clarify the type of study conducted and provide information on the sample calculation. The description of the 51 participants should be structured in a table for a clearer and more appealing presentation to the reader. Additionally, the specific and main aspects related to the study objectives should be clearly and concisely written. The clarity of the data analysis should also be improved.

  • Thank you for these suggestions. We added a short paragraph on the study design on page 4, lines 161-167, including also a concise description of the study objectives: “The current study had a cross-sectional study design. Participants from a clinical and a reference sample completed questionnaires on self-compassion and anxiety at one measurement occasion. The study objectives were to compare the level of self-compassion (compassionate and uncompassionate self-responding) between the clinical and the reference group, and the study the association between self-compassion (uncompassionate and compassionate self-responding) in both the clinical and the reference group.”

We did not do the sample size calculation a priori but only post-hoc, that’s why we included the calculation in the results section at page 10, lines 409-418, instead of in the methods section.

We now included all sociodemographic and diagnostic characteristics in Table 1 on page 5, lines 192-238.

We now described the statistical analyses in much more detail, linking the analyses to the study objectives. We specified the criteria we used for checking the assumptions, and we described the regression analyses step by step on page 6 and 7, lines 288-326. 

Results: The clarity of the main results needs improvement, and the terms and acronyms used should be reviewed.

  • We added subheadings to clarify how the results section is organized. The subheading ‘Regression analyses’ was removed and instead we used the subheadings ‘Self-compassion: clinical versus reference group’ (on page 9, line 355), ‘Self-compassion and anxiety in the clinical group’ (on page 9, line 367), and ‘Self-compassion and anxiety in the reference group’ (on page 10, line 390).

Also, we removed all acronyms and abbreviations of measures that were used and replaced them, using written out concepts, in Table 2, 3, 4 and 5.

Discussion: The authors adequately present a concise summary of the results, but it is necessary to provide plausible explanations for the observed results. The discussion should address the possible theoretical and practical implications of the results, as well as the limitations and strengths of the study and its impact on public health.

  • We have rewritten a large part of the discussion, and gave it more structure by including subheadings. As in the first version of the manuscript, we start with a summary of the results on page 10 and 11, lines 421-437. Then we gave explanations for the study findings in continuation with the research questions on page 11 and 12, lines 438-489, followed by more detailed theory including theoretical implications on page 12 and 13, lines 490-533. We made a separate paragraph on practice implications on page 13, lines 534-557. Research implications were included in the future directions, that we combined with the paragraph on the study limitations and strengths that we now also included on page 13 and 14, lines 558-593. We hope that we were able to address your feedback in the revised version of the discussion.

Conclusions: The conclusions should be modified according to the study objectives, avoiding including points that belong in the discussion section. They should focus on the strength of the study and its relevance in the research field.

  • We have now rewritten the conclusion section, hopefully in line with your suggestions. It was not easy to completely avoid all points raised in the discussion section. An alternative would be to completely discard the Conclusions section, as this section is not mandatory. The conclusions on page 14, lines 594-605 have now been formulated as follows: “This study was the first to compare a clinical group of adolescents with anxiety disorders to a group of adolescents from the general population, showing a difference in uncompassionate self-responding between the groups, showing an association between uncompassionate self-responding and anxiety in adolescents with an anxiety disorder and showing an association between compassionate and uncompassionate self-responding and anxiety in adolescents from the general population. The results of our study do not confirm the idea that compassionate and uncompassionate self-responding are two sides of the same coin. A practical implication of our study is that when self-compassion is added to treatment for adolescents with an anxiety disorder, it is important to help adolescents decrease uncompassionate self-responding such as self-judgment.“

Round 2

Reviewer 2 Report

The authors have clearly answered all my comments.